# Short Index of Job Satisfaction: Validity evidence from Portugal and Brazil

**Jorge Sinval[1,2], João Marôco[1] \***

**1** William James Center for Research, ISPA—Instituto Universitário, Lisbon, Portugal, **2** Business Research Unit (BRU-IUL), Instituto Universitário de Lisboa (ISCTE-IUL), Lisbon, Portugal

\* jpmaroco@ispa.pt

## Abstract

Job satisfaction is an important construct that is known to be associated with workers' performance and wellbeing. As such, to properly measure it, one must use adapted measures that show adequate validity evidence for the desired context. Such measures should preferably be short to allow the parsimonious use of various measures/constructs in the same data collection. The aim of this paper is to adapt the Portuguese version for Brazil and Portugal of the Short Index of Job Satisfaction (SIJS). The SIJS is a psychometric instrument that measures job satisfaction through five items. A cross-sectional study was conducted with two multi-occupational workers samples, one from Brazil ($n$ = 599) and one other from Portugal ($n$ = 572). The SIJS presented good validity evidence based on its internal structure, namely dimensionality, reliability, and measurement invariance across countries and sexes. It also revealed to be positively correlated with work engagement, and quality of work life (convergent evidence). It also has shown to be negatively associated with burnout (discriminant evidence). The SIJS showed promising validity evidence. The SIJS can be useful to be used together with other instruments, due to its small number of items, producing data with good psychometric properties.

## Introduction

Job satisfaction has a long history of proliferating organizational research [1] being one of the most–if not the most–studied variables in business science [2]. It was brought into the limelight in the 30s [3,4] being the most studied variable in the organizational behavior context [5,6]. It has various definitions. The most cited is the one provided by Locke [7], which considered job satisfaction as a positive emotional state that results from the worker's job associated experiences. Researchers can be interested in an overall measure of job satisfaction or particular job satisfaction facets, being the first usually preferred [8]. Overall job satisfaction can be seen as a formative construct, aggregating satisfaction with specific facets of the job [9]. Such approach appears as a solution to measure the overarching degree of satisfaction with various job attributes, coming with various different flavors [8].

Due to its complexity–influenced by various factors–this construct is frequently used in work and organizational studies together with other dimensions. The factors that influence job

**Data Availability Statement:** All relevant data are within the manuscript and its Supporting Information files.

**Funding:** JS and JM secured funding from the Portuguese Science Foundation (Grant number: UID/PSI/04810/2019) https://www.fct.pt/. This

work was also produced with the support of Infraestrutura Nacional de Computação Distribuída (INCD); funded by Fundação para a Ciência e a Tecnologia (FCT) and Fundo Europeu de Desenvolvimento Regional (FEDER) under the project 22153-01/SAICT/2016).

**Competing interests:** The authors have declared that no competing interests exist.

satisfaction can be considered both at the individual (mainly one's values, but also personality and mental health) and the organizational level (work, payment, promotions, peers/colleagues, supervisor, top leadership and benefits/policies) [7]. As such, a higher quality of work-life balance is expected to produce higher job satisfaction [10–12] while more job satisfaction is positively related to greater organizational commitment [13]. In the opposite direction, more stress in the workplace [14] and more surface acting [15] are related to less job satisfaction. It has been observed that workers have their job satisfaction levels decreasing parallel to the increasing tenure in a specific organization, while for people getting older–and transitioning of the organization–their satisfaction increases [16]. Lack of job satisfaction can lead to turnover [17–19], negative mood [20], reduced health and life happiness [7]. It can be related to several withdrawal behaviors [6] as absenteeism, presenteeism, and performance [21–24]. The relation with performance has long been considered the "holy grail" of satisfaction research [25] However, such relation is not unidirectional, lower performance can also lead to less job satisfaction [7]. It also has relations with several other job behaviors [26].

Despite existing many different psychometric instruments to measure the job satisfaction construct, some criticism has been made about the way this construct has been measured [27], since few of those instruments have shown satisfactory validity evidence [14]. Some of the most widespread measures are: the Index of Job Satisfaction [28] which also has a short version of five items (Short Index of Job Satisfaction [SIJS]) [29]; the Measure of Job Satisfaction (MJS) [30]; the Job in General Scale (JIG) [31] which is part of the Job Descriptive Index (JDI) [32]; the Andrew and Withey Job Satisfaction Questionnaire [33]; the Job Satisfaction Survey (JSS) [34]; the Minnesota Satisfaction Questionnaire [35]; the Michigan Organizational Assessment Questionnaire Job Satisfaction Subscale (MOAQ-JSS) [36]; and the Generic Job Satisfaction Scale (GJSS) [37]. Some of those measures are lengthy like the MJS (38 items) [30], the JIG with 18 items [31], the JSS with 36 items, or the MSQ with 100 items in the long-form, and with 20 items in the short form [35]. Other measures are shorter, like the SIJS (comprising five items), the MOAQ-JSS with three items, the Andrew and Withey Job Satisfaction with five items, or the GJSS with ten items. Van Saane et al. [14] reviewed several measures of job satisfaction and concluded that only seven of 29 reviewed measures fulfilled the authors' minimum criteria of an adequate instrument (i.e. internal consistency and validity evidence). In such a wide variety of measures, some are specific for a specific job (e.g. MJS was developed for nurses) while others can be used with multiple occupations (e.g. SIJS, JIG, JSS).

Some instruments measure job satisfaction as a unidimensional global measure, while others assume it a multidimensional construct. Examples of former are the SIJS, the JIG, the MOAQ-JSS or the Andrew and Withey Job Satisfaction Questionnaire. While the latter has as examples the MSQ, the GJSS, the JSS, or the MJS. Global measures are more parsimonious than multidimensional constructs. It is expected that the latter explains more variance, however, it might be difficult to sort out if the broader measure assures that it is measuring the construct, or if it is also measuring elements of its causal network [38]. If one aims to measure specific job satisfaction areas, the multidimensional instruments will be more useful, since they measure job satisfaction facets. Although, the global scales are useful to have a general measure of job satisfaction [31].

There are also single-item measures, which some authors find useful in specific situations [27,39] and which can be more cost-effective than multi-item measures [40]. Despite single-item measures' usefulness and correlation with scale measures–from a psychometric perspective, multiple-item measures are preferable [41,42]. Single-item measures do not allow to obtain internal consistency estimates, using them to measure psychological constructs can be seen as a fatal error [43]. As such, short measures that approach job satisfaction from a general perspective–as the SIJS–without entering in the peculiarities and the specifics of certain

occupations might be preferable to single-item measures. These short multi-item measures maintain the overall perspective of the measure with the advantage of obtaining the latent measure through the necessary multiple items. It is preferable to select a short measure that has demonstrated validity evidence in the desired population of the study, then to adapt or select some items of a longer instrument [44]. One of these short measures is the SIJS which has some advantages over other measures. It is freely available, has shown good psychometric properties and has been used in different cultures [45–48].

No studies adapting the SIJS to Portugal or Brazil were found, as such, the aim of this study is: i) adapt the SIJS in its Portuguese version for Brazil and Portugal and assess its validity evidence; and, ii) compare the job satisfaction levels between sexes and countries. Following the guidelines of the *Standards for Educational and Psychological Testing* [49] two of the possible five sources of validity evidence for psychological instruments will be analyzed. The first is based on the internal structure which refers to the reliability of the scores, dimensionality and measurement invariance. The second source of validity is based on relation to other variables. Regarding the validity evidence based on the internal structure, three hypotheses were defined. Previous studies suggest that the SIJS is a single-factor measure suitable to obtain a general job satisfaction measure, presenting satisfactory validity evidence [46,50,51]. Hypothesis 1 states that the SIJS maintains its original dimensionality (five items, one factor) supported by factorial validity evidence.

Various studies showed that the SIJS presents good reliability, namely, internal consistency estimates [52–55]. Hypothesis 2 assumes that the SIJS presents adequate evidence validity in terms of reliability of the scores.

No study was found testing measurement invariance of the SIJS between sexes or countries. Nevertheless, there is evidence that other psychological instruments used in the organizational context achieved measurement invariance among countries (i.e. Brazil and Portugal) and among sexes in the two countries [56,57]. Brazil and Portugal share more than the same language, they exchange culture and they share human capital. As such, it will be likely that the SIJS has measurement invariance between countries and sexes. Thus, Hypothesis 3 states that SIJS will hold measurement invariance (at least scalar invariance) among Brazil and Portugal, and sexes within each of the countries.

The second source of validity evidence, based on the relationships with other variables, will be checked using quality of work-life, burnout, and work engagement measures. Lack of Job satisfaction is expected to be a predictor of burnout [58–60]. On the other hand, work engagement has shown to predict job satisfaction [61–63] and by the quality of work-life [12]. Nevertheless, this source of evidence is intended to assure that job satisfaction is distinguished from associated concepts [64]. Altogether, Hypothesis 4 states that the SIJS shows adequate validity evidence based on the relation to other variables—more specifically—nomological evidence [65]. As such, it hypothesized that the SIJS will present discriminant evidence with burnout (Hypothesis 4.1), and convergent evidence with work engagement (Hypothesis 4.2) and quality of work-life (Hypothesis 4.3).

Previous research studied differences among sexes for job satisfaction among countries and sexes. The meaning attached to work seems to not differ significantly among countries [6]. However, Bozionelos and Kostopoulos [66] indicated that different countries might have differences in their job satisfaction levels. Such differences have been observed between several countries [67–69]. Pichler and Wallace [70] found that differences mainly result from composition and individual-level factors of each countries' workforce, instead of inherent cultural and historic characteristics. While, on the other hand, Hauff, Richter, and Tressin [71] indicated that some job characteristics vary significantly between countries, being partially moderated by cultural dimensions. It seems that culture might be a moderator between job

satisfaction and job characteristics [72,73]. Given that Portugal and Brazil are two countries that share language and cultural features, it is expected that job satisfaction differences between countries are small or non-existent.

In terms of sex comparisons, the theoretical arguments behind potential workplace beliefs and attitudes among women and men are related to the work-life conflict [74]. Women are historically more affected by this conflict, and consequently, they might present a lower job satisfaction than men, particularly in some life events like the first marriage or the birth of the first child [75]. Nevertheless, women can present higher job satisfaction levels despite their poor quality of work conditions, something that Hakim [76] coined as the "grateful slaves" paradox. Albeit, past research suggests that men and women are increasingly similar in terms of job and life satisfaction [77]. Altogether, Hypothesis 5 states that different job satisfaction latent mean levels are observed, among different countries (Hypothesis 5.1) and different sexes (Hypothesis 5.2).

## Method

### Sampling and data collection

For this study, two samples of multi-occupational workers were collected ($N$ = 1,171; S1 and S2 Datasets). One sample is composed of Brazilian workers ($n$ = 599) working in Brazil, and one other sample is composed of Portuguese workers ($n$ = 572) working in Portugal. The sample from Portugal had an average age of 35.83 ($SD$ = 9.76) with 37.16% being males, 83.07% had graduation or higher academic level, the occupational group with most participants was professionals (53.63%). The sample from Brazil had an average age of 35.11 ($SD$ = 10.13), with 32.77% of males, 74.39% had graduation or higher academic level, professionals were most represented occupational group (36.12%).

The socioeconomic status according to the 2014 version of the Brazilian Criteria [78] was obtained, consisting of a socioeconomic classification of the households ranging from A1 (best socioeconomic level) to E (worst socioeconomic level) (Table 1). Additional variables regarding the characterization of the samples are available in Table 1.

A non-probabilistic convenience sample was collected within a cross-sectional survey through an online tool software [80]. The inclusion criteria were being workers with a contract or formal ties with their employers, literate, and with access to a device where the survey could be fulfilled.

### Constructs and measures

**Short Index of Job Satisfaction (SIJS).**   The Index of Job Satisfaction is a self-report psychometric instrument created by Brayfield and Rothe [28]. The original version was composed of 18 items, although a shorter version with five items (SIJS) has also been proposed [29,81]. Subjects are asked to respond to each item by checking a five-point scale (1 –"Strongly Disagree", 2 –"Disagree", 3 –"Undecided", 4 –"Agree", 5 –"Strongly Agree") two of those items are reversed (Table 2). Regarding the validity evidence based on the internal structure in terms of reliability, this shorter five items version presented a good internal consistency evidence ($\alpha$ = .89) [29]. Similar evidence was found in another study ($\alpha$ = .82 to .83) [82]. *The ITC Guidelines for Translating and Adapting Tests* were used to adapt the SIJS to Portuguese in a single version (using the Portuguese Language Orthographic Agreement of 1990) that could be used both for Portugal and Brazil with the same items [83]. The pilot data on the SIJS adapted version to the Portuguese language used 15 workers from Brazil and 15 workers from Portugal.

**Quality of Working Life Scale (QWLS).**   The Quality of Work Life (i.e. second-order factor) was measured with the QWLS in its Brazil and Portugal version [84]. This instrument has

**Table 1. Demographics, career, academic level and occupational group statistics across countries.**

|  | Portugal (*n* = 572) | Brazil (*n* = 572) | Comparisons |
|---|---|---|---|
| **Academic level %** | | | |
| High school, vocational education or less | 12.26 | 12.52 | $\chi^2(4) = 153.682$; $p < .001$; $V = .395$ |
| Unfinished graduation | 4.67 | 13.09 | |
| Graduation | 29.57 | 34.16 | |
| Post-graduation (not master neither PhD) | 9.34 | 25.62 | |
| Master | 38.52 | 9.49 | |
| PhD | 5.64 | 5.12 | |
| **Demographics and career** | | | |
| Working years in the current job sector *M(SD)* | 11.23 (9.69) | 9.73 (8.61) | $t(983.35) = -2.607$; $p = .009$; $d = .164$ |
| Working years in the current organization *M(SD)* | 8.11 (8.92) | 5.84 (6.80) | $t(924.95) = -4.544$; $p < .001$; $d = .287$ |
| Working years in the current job *M(SD)* | 6.14 (7.05) | 4.97 (6.29) | $t(986.08) = -2.760$; $p = .006$; $d = .174$ |
| Sex (males) % | 37.16 | 32.77 | $\chi^2(1) = 2.024$; $p = .155$; $\varphi = .046$ |
| Age *M(SD)* | 35.83 (9.76) | 35.11 (10.13) | $t(1,039) = -1.169$; $p = .243$; $d = .072$ |
| **Occupational groups* %** | | | |
| Elementary Occupations | 1.81 | 1.55 | $\chi^2(8) = 81.394$; $p < .001$; $V = .284$ |
| Plant and Machine Operators and Assemblers | 0.60 | 0.78 | |
| Craft and Related Trades Workers | 2.22 | 2.14 | |
| Skilled Agricultural, Forestry and Fishery Workers | - | - | |
| Services and Sales Workers | 6.05 | 6.21 | |
| Clerical Support Workers | 9.48 | 27.38 | |
| Technicians and Associate Professionals | 12.90 | 8.74 | |
| Professionals | 53.63 | 36.12 | |
| Managers | 8.87 | 15.53 | |
| Armed Forces Occupations | 4.44 | 1.55 | |
| **Socioeconomic status %** | | | |
| A1 | 14.06 | 9.11 | $\chi^2(6) = 38.045$; $p < .001$; $V = .191$ |
| A2 | 54.69 | 45.16 | |
| B1 | 25.20 | 29.60 | |
| B2 | 4.69 | 10.06 | |
| C1 | 1.17 | 5.31 | |
| C2 | 0.20 | 0.38 | |
| D | - | 0.38 | |
| E | - | - | |

*Following the *International Standard Classification of Occupations ISCO-08* [79].

16 items that are answered with a seven-point scale (Likert-type; from 1- "Very Untrue" to 7 –"Very True").

This measure comprises a second-order factor–*Quality of Work Life*–and seven first-order factors (i.e. seven major needs) which are: (1) economic and family needs;(2) health and safety needs; (3) aesthetic needs; (4) actualization needs; (5) esteem needs; (6) knowledge needs; and (7) social needs [12]. This measure tries to assess the workers' perception of how the job requirements, work environment, support programs, and supervisor behavior meet their needs. In its Portuguese version for Portugal and Brazil, it has shown good psychometric properties (measurement invariance across Portugal and Brazil, and genders) [84]. It also revealed good second-order internal consistency values both for Brazil ($\omega_{L2} = .96$; $\omega_{L1} = .90$; $\omega_{partial\ L1} = .94$) and Portugal ($\omega_{L2} = .95$; $\omega_{L1} = .88$; $\omega_{partial\ L1} = .93$).

**Table 2. SIJS original and Portuguese versions.**

| Item | Original SIJS [29] | | | | | Portuguese (Brazil and Portugal) version of SIJS | | | | |
|---|---|---|---|---|---|---|---|---|---|---|
| | Strongly Disagree | Disagree | Undecided | Agree | Strongly Agree | Discordo Fortemente | Discordo | Indeciso | Concordo | Concordo Fortemente |
| | 1 | 2 | 3 | 4 | 5 | 1 | 2 | 3 | 4 | 5 |
| 1 | I feel fairly satisfied with my present job | | | | | Sinto-me razoavelmente satisfeito com meu emprego atual | | | | |
| 2 | Most days I am enthusiastic about my work | | | | | Na maioria dos dias, estou entusiasmado com o meu trabalho | | | | |
| 3[R] | Each day at work seems like it will never end (R) | | | | | Cada dia no trabalho parece não ter fim (R) | | | | |
| 4 | I find real enjoyment in my work | | | | | Sinto-me realmente satisfeito no meu trabalho | | | | |
| 5[R] | I consider my job to be rather unpleasant (R) | | | | | Considero que meu emprego é particularmente desagradável (R) | | | | |

[R]Reversed items.

**Oldenburg Burnout Inventory (OLBI).** The Oldenburg Burnout Inventory (OLBI) in its version for Portugal and Brazil [57] was used to measure burnout. Burnout is an occupational phenomenon; a workplace syndrome that results from coping unsuccessfully with chronic stressors from the occupational context [85]. This instrument assumes a second-order latent factor–Burnout–that comprises two first-order factors, Exhaustion (eight items) and Disengagement (seven items) which are rated from 5 = "Strongly Agree" to 1 = "Strongly Disagree". Disengagement refers to a distancing and gradual loss of concern with one's work recipients or contents, while exhaustion refers to a state of draining energy as a result of cognitive, emotional and physical strain [86]. The selected OLBI's version presented good validity evidence base in the internal structure (i.e. dimensionality, reliability of the scores in terms of second-order internal consistency values [$\omega_{L2}$ = .91; $\omega_{L1}$ = .86; $\omega_{partial\ L1}$ = .93], and measurement invariance among countries and sexes) and based on the relation to other variables (i.e. nomological; discriminant evidence with work engagement).

**Utrecht Work Engagement Scale (UWES).** Work engagement was measured using the Utrecht Work Engagement Scale (short-version; nine items) in its Portuguese version for Portugal and Brazil [56]. It measures work engagement as a second-order latent factor which can be defined as a fulfilling, and positive state of mind related to work, with three (first-order) dimensions: absorption, dedication, and vigor [87]. Absorption is defined as one feeling totally concentrated and happily immersed at work [88]. Dedication refers to a sense of significance, challenge inspiration, pride and enthusiasm [87]. Vigor is defined as having high levels of energy [88]. As such, each of these three dimensions has three items, which are answered using a seven-point scale, from 0 = "Never" to 6 = "Always". This instrument has presented good validity evidence with Portuguese samples [89,90], namely in terms of dimensionality, measurement invariance (among Brazil and Portugal), reliability (second-order internal consistency values [$\omega_{L2}$ = .96; $\omega_{L1}$ = .93; $\omega_{partial\ L1}$ = .97]) and expected relation to other variables [56].

## Procedure

The participants from both countries were invited to answer the selected instruments, sociodemographic and career questions. They were contacted individually or through companies that accepted to share the study by email. The cross-sectional data (i.e. cross-sectional survey) was collected using the *LimeSurvey* software [80] through the website of two major universities one in each of the two countries. The participation rate of the subjects that clicked in the invitation link from Brazil was 68.07% while 74.38% of Portuguese sample that clicked on the link fulfilled the survey. The participants were presented with the electronic informed consent in the first place, and after accepting it, had access to the survey *per se*. This study was approved by

the Committee of Ethics in Research with Human Beings (CEP) of the Faculty of Philosophy, Sciences, and Letters of Ribeirão Preto, University of São Paulo, through "platform Brazil" of CONEP (National Commission of Ethics in Research). The written consent was obtained with the approval number 33301214.2.0000.5407. This study was also approved by the Ethics Committee of the Faculty of Psychology and Education Sciences of the University of Porto, the written consent had the following statement "the Ethics Committee of the Faculty of Psychology and Education Sciences of the University of Porto, having analyzed the research project . . . considers that it respects all ethical principles and ethical standards of research and therefore gives a favorable opinion".

## Data analysis

The statistical program *R* [91] through the integrated development *RStudio* [92] was used to perform all statistical analyses. The *skimr* package [93] produced descriptive statistics. The multivariate normality was verified by the multivariate kurtosis [94] which was obtained using the *psych* package [95].

The confirmatory factor analysis (CFA) was the statistical technique selected to test the dimensionality of the instrument model. The *lavaan* package [96] was used to perform CFAs and multigroup CFA (MGCFA) with the Weighted Least Squares Means and Variances (WLSMV) estimator [97]. Some goodness-of-fit indices were selected: $\chi^2$ (chi-square), RMSEA (Root Mean Square Error of Approximation), SRMR (Standardized Root Mean Square Residual), CFI (Comparative Fit Index), NFI (Normed Fit Index), and the TLI (Tucker Lewis Index). Regarding the CFI, NFI, and TLI, Hu and Bentler [98] recommended that values above .95 should be considered as indicative of good fit. The RMSEA and SRMR values below .08 should be considered as good [99,100]. Model modifications were added through the analysis of the modification indices ($> 30$; $p < .001$) together with theoretical considerations.

The convergent evidence (in terms of internal structure) was assessed through the Average Variance Extracted [AVE; 101] calculated from polychoric correlations [102] where AVE $\geq$ .50 was considered as adequate convergent validity evidence [100].

The reliability of the scores evidence in terms of internal consistency was approached with the ordinal α [103], Composite Reliability [CR; 101] and ω [104,105] coefficients. The ordinal α coefficient was calculated from the polychoric correlations matrix. For the three coefficients, values $\geq$.70 were considered as indicative of acceptable reliability of the scores [100]. In the case of the second-order latent factors, three reliability estimates were calculated: the variance of the first-order factors explained by the second-order factor ($\omega_{L2}$), the proportion of the second-order factor explaining the total score ($\omega_{L1}$), and the proportion of variance explained by second-order factor after partialling the uniqueness of the first-order factor ($\omega_{partialL1}$). Both the ω (both for first- and second-order factors) and the α coefficients were calculated using the *semTools* package [102].

The measurement invariance of the SIJS was evaluated through multigroup confirmatory factor analysis (MGCFA) which—considering the categorical nature of items–was analyzed using the theta-parameterization [106]. Five nested models were compared to test configural invariance, metric invariance, scalar invariance, strict invariance and homogeneity of factor means. For such analysis the *semTools* package [102] was used. The full structural equation models were analyzed through the *lavaan* package [96].

Using the structural equation modeling framework, the latent variables mean scores were compared using as effect size the Cohen's *d* [107]. Additionally, for the job satisfaction raw scores the means, standard deviations and quartiles were calculated through the *doBy* package [108]. Regarding the demographics, career, academic level and occupational group variables,

the frequencies were compared using the chi-square test using the Cramér's V [109] or the φ [110] as an effect size measure using the *lsr* package [111] and the *psych* package [95] respectively. The factor means were compared using the independent two-sample *t*-test or the Welch's *t*-test when the variances were not equal. Cohen's *d* [107] was used as estimator of effect size obtained through the *lsr* package [111].

## Results

### Validity evidence based on the internal structure

**Dimensionality.** *Items' distributional properties*. The SIJS' items were analyzed in terms of its descriptive statistics (Table 3) which did not reveal severe violations in terms of univariate normality, since none of the items (for both samples) presented absolute values of Sk above 3 or absolute values of Ku above 7 [112]. In terms of multivariate normality, both samples had values of Mardia's multivariate kurtosis indicative of the absence of multivariate normality: Portugal (27.61, $p < .001$) and Brazil (16.10, $p < .001$). The maximum possible range of answers was observed in the five items and there was no outliers deletion. Altogether the distributional proprieties of the items indicate adequate psychometric sensitivity, however failing to achieve multivariate normality. As such, the WLSMV estimator that does not assume multivariate normality was used being also more adequate to the ordinal nature of the SIJS items.

*Factor related validity evidence*. The factorial validity evidence of the original proposed single latent factor was acceptable for the joint sample (Fig 1; $\chi^2(5) = 108.469$; $p < .001$; $n = 1,171$; *CFI* = .994; *NFI* = .993; *TLI* = .987; *SRMR* = .055; *RMSEA* = .133; *P*(rmsea) $\leq$ .05) < .001; 90% CI ].112; .155[). Although, after checking the modification indices it was noticed that the correlation path between the residuals of the items 3 and 5 could improve the model's fit. That correlation was added ($r = .322$; $p < .001$) and the fit to the data was very good ($\chi^2(4) = 11.228$; $p = .024$; $n = 1,171$; *CFI* = 1.000; *NFI* = .999; *TLI* = .999; *SRMR* = .021; *RMSEA* = .039; *P*(rmsea) $\leq$ .05) = .698; 90% CI ].013; .067[). All factor loadings (λ) and the items 3 and 5 residuals' correlation were statistically significant. There were no items removed. The minimum item loading for the SIJS was .57. The fit of the data to the model with one correlation between the referred items in the Brazilian sample was good ($\chi^2(4) = 11.486$; $p = .022$; $n = 599$; *CFI* = .999; *NFI* = .998; *TLI* = .997; *SRMR* = .030; *RMSEA* = .056; *P*(rmsea) $\leq$ .05) = .339; 90% CI ].019; .095[) the same was observed in the Portuguese data ($\chi^2(4) = 6.726$; $p < .001$; $n = 572$;

**Table 3. Sample 1 and 2 descriptive statistics.**

| Item | *M* | *SD* | Min | Q1 | *Mdn* | Q3 | Max | *Histogram* | Sk | Ku |
|---|---|---|---|---|---|---|---|---|---|---|
| | | | | | **Brazil** | | | | | |
| Item 1 –"*fairly satisfied*" | 3.41 | 1.27 | 1 | 3 | 4 | 4 | 5 | | -0.46 | -0.77 |
| Item 2 – "enthusiastic" | 3.49 | 1.19 | 1 | 3 | 4 | 4 | 5 | | -0.43 | -0.74 |
| Item 3[R] – "it will never end" | 3.54 | 1.25 | 1 | 3 | 4 | 5 | 5 | | -0.58 | -0.66 |
| Item 4 – "real enjoyment" | 3.40 | 1.23 | 1 | 3 | 4 | 4 | 5 | | -0.43 | -0.77 |
| Item 5[R] –"unpleasant" | 4.15 | 1.12 | 1 | 3 | 5 | 5 | 5 | | -1.13 | 0.28 |
| | | | | | **Portugal** | | | | | |
| Item 1 – "fairly satisfied" | 3.48 | 1.11 | 1 | 3 | 4 | 4 | 5 | | -0.49 | -0.42 |
| Item 2 – "enthusiastic" | 3.51 | 1.13 | 1 | 3 | 4 | 4 | 5 | | -0.56 | -0.38 |
| Item 3[R] – "it will never end" | 3.68 | 1.13 | 1 | 3 | 4 | 5 | 5 | | -0.60 | -0.42 |
| Item 4 – "real enjoyment" | 3.31 | 1.16 | 1 | 3 | 3 | 4 | 5 | | -0.40 | -0.64 |
| Item 5[R] –"unpleasant" | 4.30 | 1.01 | 1 | 4 | 5 | 5 | 5 | | -1.45 | 1.51 |

[R]Reversed items; the descriptive statistics of such items refer to the recoded scores.

$CFI = 1.000$; $NFI = 1.000$; $TLI = 1.000$; $SRMR = .019$; $RMSEA = .035$; $P(\text{rmsea}) \leq .05) = .661$; 90% CI ].000; .078[.

*Convergent validity evidence.* The AVE value for the job satisfaction factor was .58, which is indicative of good convergent validity evidence for this factor. Good AVE values were also observed for each country ($AVE_{Brazil} = .54$; $AVE_{Portugal} = .65$). Globally, these results indicate good convergent validity evidence in terms of internal structure for SIJS. In other words, the items contained in the job satisfaction factor relate to each other.

## Reliability of the scores

The reliability of the scores was assessed through internal consistency estimates which revealed to be globally very good values for both samples (Table 4). The values of the internal consistency estimates were very good ($\geq .84$) for the Brazilian data, with the CR obtaining the highest value. In the case of the Portuguese sample all internal consistency estimates were also very good ($\geq .88$). Such results are indicative of very good evidence in terms of reliability, suggesting that items' scores are homogenously crowded together around the job satisfaction latent variable.

## Measurement invariance

The measurement invariance among countries and sexes was tested through a group of nested models with increasing constraints (Table 5). Scalar invariance was achieved among countries considering the $\Delta CFI < .010$ criterion [113] while using the $\Delta\chi^2$ criterion [114] only metric invariance was achieved. Regarding the measurement invariance among sexes in the Brazilian sample, full uniqueness measurement invariance was obtained using the $\Delta CFI < .010$ criterion, while the $\Delta\chi^2$ criterion only supported metric invariance. The Portuguese sample showed that full uniqueness measurement invariance was supported by both criteria (i.e. $\Delta CFI < .010$ and $\Delta\chi^2$). That is to say that the SIJS measures job satisfaction in the same way across countries and sexes, allowing to interpret the differences in terms of group differences in the job satisfaction construct.

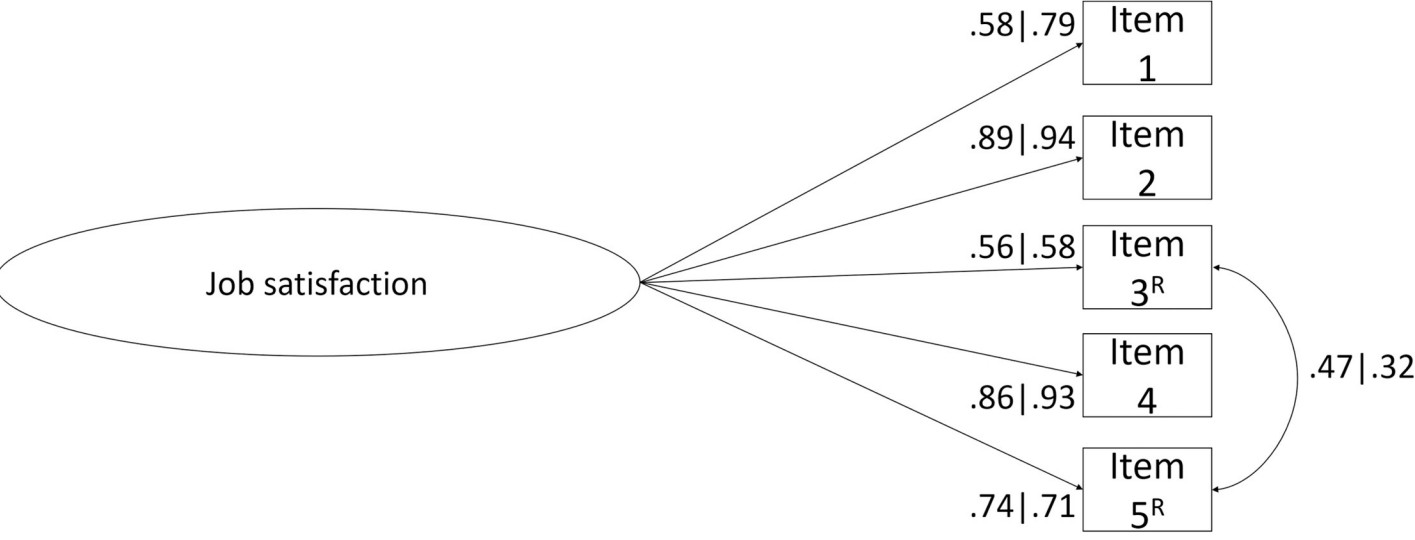

**Fig 1. SIJS latent structure (five items).** Factor loadings for each item are shown (Brazil | Portugal). [R]Reversed items.

**Table 4. Internal consistency estimates for both samples.**

| SIJS dimension | SIJS—5 items | | | | | | | | |
|---|---|---|---|---|---|---|---|---|---|
| | Brazil | | | Portugal | | | Total | | |
| | α | ω | CR | α | ω | CR | α | ω | CR |
| Job Satisfaction | .86 | .84 | .87 | .90 | .88 | .90 | .88 | .86 | .88 |

## Validity evidence based on relations to other variables

Three additional measures were used to investigate the validity evidence based on relations to other variables, particularly the nomological validity both in terms of convergent and discriminant evidence [49]. Using the OLBI second-order factor model which revealed an acceptable fit to the data ($\chi^2$(86) = 1,039.970; $p < .001$; $n = 1,109$; $CFI = .984$; $NFI = .983$; $TLI = .981$; $SRMR = .065$; $RMSEA = .100$; $P(\text{rmsea}) \leq .05) < .001$; 90% CI ].095; .106[) after adding three pairs of correlations between items' residuals (i.e. $Item1_{OLBI}$—$Item7_{OLBI}$; $Item10_{OLBI}$–$Item12_{OLBI}$; and $Item4_{OLBI}$–$Item16_{OLBI}$). Such correlations were added after the analysis of modification indices because the items in each pair belong to the same factor. Since it seems plausible that residuals of the same factor are correlated [115]. The UWES-9 second-order model also obtained an acceptable fit ($\chi^2$(25) = 437.948; $p < .001$; $n = 1,100$; $CFI = .998$; $NFI = .998$; $TLI = .997$; $SRMR = .041$; $RMSEA = .123$; $P(\text{rmsea}) \leq .05) < .001$; 90% CI ].113; .133[) after adding a constraint to the variance of one of the first-order factors–*Dedication*–fixing it to 0.001 to avoid negative variance. The QWLS-16 presented good fit ($\chi^2$(97) = 931.475; $p < .001$; $n = 1,108$; $CFI = .991$; $NFI = .990$; $TLI = .988$; $SRMR = .064$; $RMSEA = .088$; $P(\text{rmsea}) \leq .05) < .001$; 90% CI ].083; .093[). Both in terms of first- and second-order factors the reliability

**Table 5. Measurement invariance between countries.**

| Model | $\chi^2$ | df | CFI scaled | $\Delta\chi^2$ | $\Delta$CFI scaled |
|---|---|---|---|---|---|
| **Countries** | | | | | |
| Configural | 106.14 | 10 | .981 | - | - |
| Metric | 113.33 | 14 | .985 | $7.820^{ns}$ | -.004 |
| Scalar | 149.57 | 28 | .982 | $49.056^{***}$ | .003 |
| Full uniqueness | 333.08 | 33 | .959 | $233.143^{***}$ | .023 |
| Latent means | 346.43 | 34 | .975 | $2.125^{ns}$ | -.016 |
| **Sex Brazil** | | | | | |
| Configural | 76.147 | 10 | .955 | - | - |
| Metric | 82.685 | 14 | .962 | $6.021^{ns}$ | -.007 |
| Scalar | 92.337 | 28 | .963 | $14.871^{ns}$ | -.001 |
| Full uniqueness | 102.484 | 33 | .962 | $11.675^{*}$ | .001 |
| Latent means | 107.401 | 34 | .975 | $0.863^{ns}$ | -.013 |
| **Sex Portugal** | | | | | |
| Configural | 27.010 | 10 | .993 | - | - |
| Metric | 27.468 | 14 | .994 | $1.260^{ns}$ | .001 |
| Scalar | 41.714 | 28 | .994 | $20.350^{ns}$ | .000 |
| Full uniqueness | 45.361 | 33 | .994 | $5.572^{ns}$ | .000 |
| Latent means | 50.736 | 34 | .997 | $0.863^{ns}$ | -.003 |

$^{ns}p > .05$

$^{*}p \leq .05$

$^{***}p < .001$.

of the scores in terms of internal consistency for QWLS, UWES-9 and OLBI presented acceptable to very good evidence (Table 6).

In nomological terms, the convergent evidence was good, since both the model of quality of work life correlating with job satisfaction ($\chi^2$(180) = 1,126.605; $p < .001$; $n = 1,108$; $CFI = .993$; $NFI = .992$; $TLI = .992$; $SRMR = .054$; $RMSEA = .069$; $P$(rmsea) $\leq .05$) $< .001$, 90% CI ].065; .073[) and the model of work engagement correlating with job satisfaction ($\chi^2$(73) = 669.163; $p < .001$; $n = 1,100$; $CFI = .998$; $NFI = .997$; $TLI = .997$; $SRMR = .038$; $RMSEA = .086$; $P$(rmsea) $\leq .05$) $< .001$, 90% CI ].080; .092[) presented significant and strong correlations ($r_{JS, QWL} = 0.82$; $r_{JS, WE} = 0.86$, respectively).

The model where job satisfaction was correlated with burnout showed good fit to the data ($\chi^2$(165) = 1,558.286; $p < .001$; $n = 1,109$; $CFI = .988$; $NFI = .987$; $TLI = .986$; $SRMR = .061$; $RMSEA = .087$; $P$(rmsea) $\leq .05$) $< .001$, 90% CI ].083; .091[). The hypothetical model indicates a large and negative correlation between job satisfaction and burnout ($r_{JS, B} = -0.90$; $p < .001$). Such result suggests good discriminant evidence in nomological terms. Altogether, the SIJS' validity evidence based on relations to other variables was satisfactory, showing associations in the expected directions in other associated measures.

## Job satisfaction comparisons among sex and country

The job satisfaction latent means were compared through the structural equation modeling framework. The comparison among countries did not reveal statistically significant differences ($\Delta\chi^2_{scaled}$(1) = 2.125; $p = 0.145$; $d = 0.086$). Also, the comparison among sexes did not reveal statistically significant differences ($\Delta\chi^2_{scaled}$(1) = 0.126 $p = 0.723$; $d = -0.003$). For potential comparison with other studies, job satisfaction means, standard deviations, and quartiles of the raw scores per country and sex are provided (Table 7).

## Discussion

The SIJS presented a promising fit to data from multi-occupational workers from Portugal and Brazil. The minimum λ value was high, and the instrument presented acceptable convergent validity evidence. Globally, the SIJS presented acceptable values of internal consistency, with scalar invariance obtained among the Portuguese and Brazilian samples and sexes. This is in line with the good psychometric properties observed in different countries such as the USA [82], Italy [116], New Zealand [47] or the UAE [52]. Due to its good validity evidence, the SIJS has been reported as a commonly used instrument of job satisfaction in studies published in top journals [117].

The dimensionality of the SIJS was successfully confirmed (H1), presenting an excellent fit to the data. However, a modification was added to the structure: the correlation among items' 3 and 5 residuals. Such correlation can be justified since both items belong to the same latent factor, and additionally both items are reversed. Interestingly, a study by Jabeen et al. [52] revealed problems with this pair of reversed items (i.e. unexpected factor loadings), which lead the authors to remove those items from the analysis. The discussion about the inclusion of reversed items reveals that there are specific trade-offs that researchers should be aware of [118–120].

The reliability evidence confirmed to be good (H2). Such findings are corroborated by other studies where the SIJS revealed adequate to good reliability evidence in terms of internal consistency [45–48]. Our study's internal consistency estimates values were good in the three different used estimates (i.e. α, ω and CR).

Regarding Hypothesis 3, it was also confirmed. A minimum of scalar invariance was achieved both for countries and sexes. As far as we know, none of the other studies that used

**Table 6. UWES-9, OLBI, and QLWS internal consistency estimates.**

| Estimate | UWES-9[2L] | OLBI[2L] | QWLS[2L] |
|---|---|---|---|
| Internal consistency estimates | | | |
| ω | Vigor = .94 | Disengagement = .68 | Health & Safety = .43 |
| | | | Economic & Family = .64 |
| | Dedication = .91 | | Social = .46 |
| | | | Esteem = .77 |
| | | Exhaustion = .86 | Actualization = .91 |
| | Absorption = .88 | | Knowledge = .90 |
| | | | Aesthetics = .83 |
| $\omega_{partial\ L1}$ | .97 | .93 | .96 |
| $\omega_{L1}$ | .93 | .86 | .92 |
| $\omega_{L2}$ | .96 | .92 | .96 |

[2L]Second-order latent version.

the SIJS tested the measurement invariance of this instrument. As such, this finding is an interesting contribution to the literature. Strong measurement (scalar) invariance is an essential condition to directly compare an instrument score among groups [121].

The validity evidence based on the relation to other variables revealed to be satisfactory, since the tested full structural equation models presented paths that occurred as expected, confirming Hypothesis 4. More specifically, job satisfaction was associated with burnout (H4.1). The burnout association with overall job satisfaction is well known, albeit some argue that they can have bidirectional relations [122]. Others [e.g., 123] reported a strong negative correlation between emotional exhaustion (burnout's core dimension) and job satisfaction. Emotional exhaustion has been also observed to explain specific types of job satisfaction (e.g. satisfaction with the communication) [124].

Job satisfaction also correlated with work engagement (H4.2) a finding that has parallel in other studies [125,126]. Conversely in other studies, some authors tested work engagement as a predictor of job satisfaction, finding also a positive relationship between both variables [127,128].

The relation with quality of work-life was successfully verified (H4.3) similarly to previous research which showed that quality of work-life predicts job satisfaction [12,52]. The correlation magnitude is large which is somehow expected, since the used quality of work-life measure approaches the perception of needs satisfaction related to the supervisory behavior, job requirements, and work environment which–when occurs–conducts to job satisfaction [12].

**Table 7. Job satisfaction raw scores: Quartiles, means, and standard deviations for countries and sexes.**

| | | | | | | | | | | | | | | |
|---|---|---|---|---|---|---|---|---|---|---|---|---|---|---|
| **Brazil** | | | | | | | | | | | | | | |
| **Female (n = 355)** | | | | | **Male (n = 173)** | | | | | **Total (n = 599)**[*] | | | | |
| *M* | *SD* | $P_{25}$ | $P_{50}$ | $P_{75}$ | *M* | *SD* | $P_{25}$ | $P_{50}$ | $P_{75}$ | *M* | *SD* | $P_{25}$ | $P_{50}$ | $P_{75}$ |
| 3.58 | 0.95 | 3.00 | 3.80 | 4.20 | 3.68 | 0.87 | 3.00 | 3.80 | 4.20 | 3.60 | 0.92 | 3.00 | 3.80 | 4.20 |
| **Portugal** | | | | | | | | | | | | | | |
| **Female (n = 323)** | | | | | **Male (n = 191)** | | | | | **Total (n = 572)**[*] | | | | |
| 3.68 | 0.91 | 3.20 | 3.80 | 4.40 | 3.61 | 0.89 | 3.00 | 3.80 | 4.20 | 3.66 | 0.90 | 3.00 | 3.80 | 4.40 |

[*]Some subjects did not inform their Sex.

Portugal and Brazil did not present statistically significant differences regarding job satisfaction (H5.1). Since the sample was mainly constituted by workers with higher education, such finding was plausible. Others [e.g., 129], found that job satisfaction was rather similar across various countries among the higher education graduates. Countries with different cultural backgrounds can display different perceptions of job satisfaction, and such potential differences should be controlled before making comparisons [130]. Something that—regarding the Brazil and Portugal comparison—was guaranteed both by the procedure of adaptation of the items which considered both countries' cultural specificities and through the verification of the scalar invariance of the SIJS. Only after such evidence, direct comparisons of the scores can be established [131].

The hypothesis tested in H5.2 was not supported by the data. In Brazil, a previous study among health-care workers did not find statistically significant differences among genders [132]. In Portugal there are mixed findings, with some studies not finding statistically significant differences among men and women [133,134] while others reported that men had higher job satisfaction than women[135]. Past studies in other countries found differences between sexes both with males having greater job satisfaction after controlling for similar job ranks [136], or the reverse effect [137], or even the absence of differences between women and men [138,139]. Kaiser [135] found that job satisfaction differences among sexes are expected to be insignificant within a higher job status segment. This seems a plausible explanation since the collected samples in the present paper have high frequencies of higher academic levels, which are expected to be associated with higher job status. Zou [140] reported that when work orientations are taken into account, job satisfaction differences among women and men are eliminated. Also, there is evidence that women and men are similar in the crossover and spillover dynamics regarding job satisfaction, job insecurity, own family and partner's family life satisfaction [141]. Various explanations exist for such absence, or even the possible greater job satisfaction perceived by women [142] which would be seen as a job satisfaction paradox [135]. Nulling effect of covariate variables of sex, like education, age and paygrade can explain the absence of differences [143],. perhaps due to the evolution of the demographics and women's attitudes towards work [77]. Such findings can be also explained by the less apparent differences among sexes in terms of job roles, qualifications, earnings and other indicators [144,145]. Such a fading-out process might be explained by appropriate labor market interventions by the governing institutions [135]. Hodson [142] suggested that women can report equal or greater job satisfaction than men, despite women's worst job conditions because of two possible processes that may operate in conjunction. First, women can use different comparison groups. Second, men can be more inclined to report situations of dissatisfaction than women.

## Conclusions

The short measure (i.e. SIJS) can be taken as a good measure, since it has been used successfully in several studies [e.g. 6], and it also demonstrated promising validity in this study. Despite being shorter than several other instruments versions that measure job satisfaction [146] the SIJS revealed very promising validity evidence, both based in the internal structure as also based on the relations to other variables. When a researcher wants to use several measures, the length of the selected instruments is a fundamental issue to avoid respondents' fatigue. As such, having short scales that show good validity evidence is crucial since lengthy surveys can lead to fatigue and consequently have increased insufficient effort responding [147,148].

Some limitations of the current study should be mentioned. The sample was a convenience sample (i.e. non-probabilistic). Desirably in future studies probability sampling techniques

should be used. Additionally, the two current samples do not have the same proportions of occupational groups, sex and academic level in each country. And since, the conclusions regarding invariance should be analyzed with caution. In this cross-sectional study, no information can be extracted regarding the temporal stability of the SIJS structure. In future research, researchers might address this issue by testing longitudinal measurement invariance [149]. Research with other samples (i.e. more specific occupational groups) is expected to occur in future studies, as also in other countries/cultures, to approach the transcultural invariance robustness of the SIJS.

## Supporting information

**S1 Dataset. The dataset used.**
(CSV)

**S2 Dataset. The dataset codebook.**
(CSV)

## Acknowledgments

The authors would like to thank the Portuguese national occupational health program of the Directorate-General of Health (DGS). We thank Dr. Edwin A. Locke (Robert H. Smith School of Business [Emeritus], University of Maryland, College Park, MD, United States of America) for helpful comments.

## Author Contributions

**Conceptualization:** Jorge Sinval, João Marôco.

**Data curation:** Jorge Sinval.

**Formal analysis:** Jorge Sinval.

**Funding acquisition:** João Marôco.

**Investigation:** Jorge Sinval.

**Methodology:** Jorge Sinval, João Marôco.

**Project administration:** Jorge Sinval.

**Software:** Jorge Sinval.

**Supervision:** João Marôco.

**Visualization:** Jorge Sinval.

**Writing – original draft:** Jorge Sinval.

**Writing – review & editing:** Jorge Sinval, João Marôco.

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
