## [Decision Letter · Decision Letter 0]

26 Nov 2019

PONE-D-19-28151

Short Index of Job Satisfaction: Validity Evidence from Portugal and Brazil

PLOS ONE

Dear Dr Marôco,

Thank you for submitting your manuscript to PLOS ONE. After careful consideration, we feel that it has merit but does not fully meet PLOS ONE’s publication criteria as it currently stands. Therefore, we invite you to submit a revised version of the manuscript that addresses the points raised during the review process.

Your paper has been commented by three reviewers that are acknowledged academics in the topic addressed in the study. You will find that two of them see potential in the paper but recommend a considerable set of revisions from you, and one of the reviewers advised against its consideration for being published in PLOS ONE. As, personally, I see this study as valuable and probably useful for occupational researchers worldwide, I decided to give you the opportunity to revise and resubmit an improved version of the manuscript, as long as all the received comments be properly addressed, the changes/amendments done properly detailed and explained, and the contents considerably updated, in accordance to the suggestions provided by our reviewers. Please also consider to put a bigger effort on the presentation of the paper for non-expert readers.

We would appreciate receiving your revised manuscript by Jan 10 2020 11:59PM. To enhance the reproducibility of your results, we recommend that if applicable you deposit your laboratory protocols in protocols.io, where a protocol can be assigned its own identifier (DOI) such that it can be cited independently in the future. For instructions see: http://journals.plos.org/plosone/s/submission-guidelines#loc-laboratory-protocols

We look forward to receiving your revised manuscript.

Kind regards,

Sergio A. Useche, Ph.D.

Academic Editor

PLOS ONE

Journal Requirements:

3. Thank you for including your ethics statement: "Human Subject Research (involving human participants)

Brazil data:

Name: Committee of Ethics in Research with Human Beings (CEP) of the Faculty of Philosophy, Sciences and Letters of Ribeirão Preto, University of São Paulo, through “platform Brazil” of CONEP (National Commission of Ethics in Research).

Approval number: CAAE no. 33301214.2.0000.5407

Form of consent obtained: written

Portugal data:

Name: Ethics Committee of the Faculty of Psychology and Education Sciences of the University of Porto (on

Statement indicating approval of this research: "A Comissão de Ética da Faculdade de Psicologia e de Ciências da Educação da

Universidade do Porto, tendo analisado o projeto de investigação de Jorge Fernando Pereira Sinval, intitulado “Saúde Ocupacional: metanálise, modelos e instrumentos”, orientado na FPCEUP pela Prof.ª Cristina Queirós, considera que ele respeita todos os princípios éticos e normas deontológicas da investigação pelo que emite um parecer favorável à sua realização."

Form of consent obtained: written"

Please amend your current ethics statement to confirm that your named institutional review boards / ethics committees specifically approved this study.

The authors would like to thank the Portuguese national occupational health program of the

Directorate-General of Health (DGS). This work was produced with the support of INCD funded by

FCT and FEDER under the project 22153-01/SAICT/2016. We thank to Dr. Edwin A. Locke (Robert

H. Smith School of Business [Emeritus], University of Maryland, College Park, MD, United States of

America) for helpful comments.

Authors: JS and JM

Grant number: UID/PSI/04810/2019

Full name of the funder: Portuguese Science Foundation

URL of funder website: https://www.fct.pt/

Reviewers' comments:

Reviewer's Responses to Questions

**Comments to the Author**

1. Is the manuscript technically sound, and do the data support the conclusions?

Reviewer #1: Yes

Reviewer #2: Yes

Reviewer #3: Yes

2. Has the statistical analysis been performed appropriately and rigorously? 

Reviewer #1: Yes

Reviewer #2: Yes

Reviewer #3: Yes

3. Have the authors made all data underlying the findings in their manuscript fully available?

Reviewer #1: Yes

Reviewer #2: No

Reviewer #3: No

4. Is the manuscript presented in an intelligible fashion and written in standard English?

Reviewer #1: Yes

Reviewer #2: No

Reviewer #3: Yes

5. Review Comments to the Author

Reviewer #1: The paper looks more like an exercise in statistical analysis as the scale is only a five item scale and it should work as a uni-dimensional scale. Thus the bigger question is what can we learn for the entire exercise.

Reviewer #2: Main claims of the paper

The authors aim to validate the SIJS in a new version in Portuguese in two samples in Brazil and Portugal. This has been done via convenience samples in both countries.

The manuscript is well organized and has a clear presentation of the background and the ideas behind.

Introduction, hypotheses

The introduction is clear and thorough and the development of the hypotheses is comprehensible. Regarding the hypotheses for sex and countries, these should be restated. Both hypotheses in H5 are formulated as null-hypotheses which cannot be proven: “Altogether, similar job satisfaction latent mean levels are expected to be observed (Hypothesis 5) among different countries (Hypothesis 5.1) and different sexes (Hypothesis 5.2).”

This point also can be seen in the results and discussion, see: “Portugal and Brazil showed no differences regarding job satisfaction, which supported H5.1.”. Again, “no differences” cannot be proven, so this topic should be considered. The same as for sexes is for H 5.2 in the comparison between the countries and here the authors themselves write about mixed results (p. 21), which again points to the principal problem of stating and trying to prove a null-hypothesis.

Methods

The data set does not provide information for occupation, school level, socioeconomic level. If PLOS needs the complete data set then this information should be added.

The description of the data set not satisfying for re-analyses as e.g. the scale information is lacking (which is understandable as this is not possible in an xls-file in comparison to an SPSS-file).

The internal consistency values for the instruments for the previous versions should be included in the methods section.

Data and analyses

As minor point, in “Table 3. Sample 1 and 2 Descriptive Statistics.” It would be helpful to have at least an abbreviation of the item content.

Discussion

The discussion follows the argumentation line, refers to the literature and research findings and discusses the results in this context.

Minor points

The language should be checked. E.g. the following sentence is not fully clear: „The meaning attached to work seems that not differs significantly among countries (Judge et al., 2001), as so it expected that job satisfaction differences between countries are not so great or inexistent.“

Some other examples.

„As such, each of these three dimensions was three items“

Reviewer #3: The main aim of this paper was to adapt the Portuguese version for Brazil and Portugal of the Short Index of Job Satisfaction (SIJS) in two different population in a cross-sectional manner. The authors have tested the convergent and discriminant validity and sex differences in SIJS in both countries. The results seem to indicate a good psychometric properties but some concerns remained before publishing the article, which are listed below.

1. Readers may feel difficult to understand what is SIJS in the abstract. Please add some more information about it for readability.

2. The rationale for analyzing sex differences seems weak to persuade the reviewer.

3. I don’t see how the sample was selected. How many was recruited and what was the response rate?

4. Table 1 doesn’t show the statistical differences between Portugal and Brazilian sample.

5. Introduction is well-written but the rationale for selecting SIJS among other measures is weak.

6. Table 1: Please explain socioeconomic status under table 1.

7. Was translation of English version of SIJS to Portuguese was back translated to English to check the proper meaning?

6. PLOS authors have the option to publish the peer review history of their article (what does this mean?). If published, this will include your full peer review and any attached files.

Reviewer #1: No

Reviewer #2: No

Reviewer #3: Yes: Akinori Nakata

---

## [Author Response · Author response to Decision Letter 0]

16 Dec 2019

REVISION NOTES

“Short Index of Job Satisfaction: Validity Evidence from Portugal and Brazil”

MS# PONE-D-19-28151

Thank you for reviewers’ comments. We found the comments very helpful and made a sincere effort to revise the paper based on the feedback from the reviewers. 

Editor

Editor’s comments

Your paper has been commented by three reviewers that are acknowledged academics in the topic addressed in the study. You will find that two of them see potential in the paper but recommend a considerable set of revisions from you, and one of the reviewers advised against its consideration for being published in PLOS ONE. As, personally, I see this study as valuable and probably useful for occupational researchers worldwide, I decided to give you the opportunity to revise and resubmit an improved version of the manuscript, as long as all the received comments be properly addressed, the changes/amendments done properly detailed and explained, and the contents considerably updated, in accordance to the suggestions provided by our reviewers. Please also consider to put a bigger effort on the presentation of the paper for non-expert readers.

Authors’ response 

We tried to make the paper more presentable for non-expert readers.

 

Reviewer 1

Reviewer 1’s comments

The paper looks more like an exercise in statistical analysis as the scale is only a five item scale and it should work as a uni-dimensional scale. Thus the bigger question is what can we learn for the entire exercise.

Authors’ response

This is a psychometric manuscript, where the SIJS Portuguese version for Portugal and Brazil is adapted and deeply evaluated in its psychometric properties. It is a useful measure for various reasons, it is one of the most measured constructs in the organizational contexts, this particular measure (i.e. SIJS) is widely used, it is free for use, there is no published study evaluating the psychometric properties of this instrument, it showed strong psychometric properties (e.g. measurement invariance among sexes and countries).

Reviewer 2

Reviewer 2’s comments

1 - The authors aim to validate the SIJS in a new version in Portuguese in two samples in Brazil and Portugal. This has been done via convenience samples in both countries. The manuscript is well organized and has a clear presentation of the background and the ideas behind.

2 - Introduction, hypotheses

The introduction is clear and thorough and the development of the hypotheses is comprehensible. Regarding the hypotheses for sex and countries, these should be restated. Both hypotheses in H5 are formulated as null-hypotheses which cannot be proven: “Altogether, similar job satisfaction latent mean levels are expected to be observed (Hypothesis 5) among different countries (Hypothesis 5.1) and different sexes (Hypothesis 5.2).”

This point also can be seen in the results and discussion, see: “Portugal and Brazil showed no differences regarding job satisfaction, which supported H5.1.”. Again, “no differences” cannot be proven, so this topic should be considered. The same as for sexes is for H 5.2 in the comparison between the countries and here the authors themselves write about mixed results (p. 21), which again points to the principal problem of stating and trying to prove a null-hypothesis.

3- Methods

The data set does not provide information for occupation, school level, socioeconomic level. If PLOS needs the complete data set then this information should be added.

The description of the data set not satisfying for re-analyses as e.g. the scale information is lacking (which is understandable as this is not possible in an xls-file in comparison to an SPSS-file).

The internal consistency values for the instruments for the previous versions should be included in the methods section.

4 - Data and analyses

As minor point, in “Table 3. Sample 1 and 2 Descriptive Statistics.” It would be helpful to have at least an abbreviation of the item content.

5 - Discussion

The discussion follows the argumentation line, refers to the literature and research findings and discusses the results in this context.

6 - Minor points

The language should be checked. E.g. the following sentence is not fully clear: „The meaning attached to work seems that not differs significantly among countries (Judge et al., 2001), as so it expected that job satisfaction differences between countries are not so great or inexistent.“

Some other examples.

„As such, each of these three dimensions was three items“

Authors’ response

1 - Thank you.

2 - The hypotheses were reformulated as per the reviewer’s suggestion.

3 - The occupational group, academic level, and socioeconomic status were added to the dataset (i.e. S1_Dataset.csv).

Unfortunately, csv files do not allow to added labels to variables or values.

The internal consistency values for the instruments previous versions were added.

4 -The items’ content is presented in Table 2. However, if the reviewer finds it strictly necessary, the items content can be repeated in this table.

5 - Thank you.

6 - We carefully reviewed the writing of the manuscript.

 

Reviewer 3

Reviewer 3’s comments

The main aim of this paper was to adapt the Portuguese version for Brazil and Portugal of the Short Index of Job Satisfaction (SIJS) in two different population in a cross-sectional manner. The authors have tested the convergent and discriminant validity and sex differences in SIJS in both countries. The results seem to indicate a good psychometric properties but some concerns remained before publishing the article, which are listed below.

1. Readers may feel difficult to understand what is SIJS in the abstract. Please add some more information about it for readability.

2. The rationale for analyzing sex differences seems weak to persuade the reviewer.

3. I don’t see how the sample was selected. How many was recruited and what was the response rate?

4. Table 1 doesn’t show the statistical differences between Portugal and Brazilian sample.

5. Introduction is well-written but the rationale for selecting SIJS among other measures is weak.

6. Table 1: Please explain socioeconomic status under table 1.

7. Was translation of English version of SIJS to Portuguese was back translated to English to check the proper meaning?

Authors’ response

Thank you.

1. Information about the SIJS was added to the abstract.

2. The rationale for analyzing sex differences was improved.

3. Information about how the sample was obtained was added, and the response rates from the participants in this study (see Procedures section).

4. The statistical tests of the frequencies and means comparisons were added.

5. The SIJS is freely available, it’s a short easy to apply measure, and it shows good psychometrics in several countries. This information was added to the introduction.

6. The socioeconomic status was evaluated according to the the “Critério Brasil” (Associação Brasileira de Empresas de Pesquisa 2012) which consists in a standard socioeconomic status criteria.

7. Yes, it was back-translated as suggested by the ITC guidelines (International Test Commission 2018) (see subsection “Short Index of Job Satisfaction (SIJS)” inside the section “Constructs and measures”).

References

Associação Brasileira de Empresas de Pesquisa. (2012). Critério de classificação econômica do Brasil [Criteria for economic classification in Brazil]. http://www.abep.org/

International Test Commission. (2018). ITC Guidelines for Translating and Adapting Tests (second edition). International Journal of Testing, 18(2), 101–134. https://doi.org/10.1080/15305058.2017.1398166

---

## [Decision Letter · Decision Letter 1]

24 Feb 2020

PONE-D-19-28151R1

Short Index of Job Satisfaction: Validity Evidence from Portugal and Brazil

PLOS ONE

Dear Dr Marôco,

Thank you for submitting your manuscript to PLOS ONE. After careful consideration, we feel that it has merit but does not fully meet PLOS ONE’s publication criteria as it currently stands. Therefore, we invite you to submit a revised version of the manuscript that addresses the points raised during the review process.

The revised version of your paper has been scientifically judged by three reviewers. Overall, they all agree on the potential of the paper and its value for occupational and organizational research. Furthermore, two of them consider that their previous comments have been adequately responded, and only one of the reviewers asked for some amendments, most of them minor. Please carefully check and amend/clarify these points, including the response to the reviewer' queries and the detail of the changes made in your rebuttal letter.

We would appreciate receiving your revised manuscript by Apr 09 2020 11:59PM. To enhance the reproducibility of your results, we recommend that if applicable you deposit your laboratory protocols in protocols.io, where a protocol can be assigned its own identifier (DOI) such that it can be cited independently in the future. For instructions see: http://journals.plos.org/plosone/s/submission-guidelines#loc-laboratory-protocols

We look forward to receiving your revised manuscript.

Kind regards,

Sergio A. Useche, Ph.D.

Academic Editor

PLOS ONE

Reviewers' comments:

Reviewer's Responses to Questions

**Comments to the Author**

1. If the authors have adequately addressed your comments raised in a previous round of review and you feel that this manuscript is now acceptable for publication, you may indicate that here to bypass the “Comments to the Author” section, enter your conflict of interest statement in the “Confidential to Editor” section, and submit your "Accept" recommendation.

Reviewer #1: All comments have been addressed

Reviewer #2: (No Response)

Reviewer #3: All comments have been addressed

2. Is the manuscript technically sound, and do the data support the conclusions?

Reviewer #1: Yes

Reviewer #2: Yes

Reviewer #3: Yes

3. Has the statistical analysis been performed appropriately and rigorously? 

Reviewer #1: Yes

Reviewer #2: Yes

Reviewer #3: Yes

4. Have the authors made all data underlying the findings in their manuscript fully available?

Reviewer #1: Yes

Reviewer #2: No

Reviewer #3: Yes

5. Is the manuscript presented in an intelligible fashion and written in standard English?

Reviewer #1: Yes

Reviewer #2: Yes

Reviewer #3: Yes

6. Review Comments to the Author

Reviewer #1: The authors have carried out the necessary revisions and the paper looks better than the original thus more acceptable.

Reviewer #2: Main claims of the paper

1 The authors aim to validate the SIJS in a new version in Portuguese in two samples in Brazil and Portugal. This has been done via convenience samples in both countries.

The manuscript is well organized and has a clear presentation of the background and the ideas behind.

Answer Authors

1 - Thank you.

2 Introduction, hypotheses

The introduction is clear and thorough and the development of the hypotheses is comprehensible. Regarding the hypotheses for sex and countries, these should be restated. Both hypotheses in H5 are formulated as null-hypotheses which cannot be proven: “Altogether, similar job satisfaction latent mean levels are expected to be observed (Hypothesis 5) among different countries (Hypothesis 5.1) and different sexes (Hypothesis 5.2).”

This point also can be seen in the results and discussion, see: “Portugal and Brazil showed no differences regarding job satisfaction, which supported H5.1.”. Again, “no differences” cannot be proven, so this topic should be considered. The same as for sexes is for H 5.2 in the comparison between the countries and here the authors themselves write about mixed results (p. 21), which again points to the principal problem of stating and trying to prove a null-hypothesis.

Answer Authors

2 - The hypotheses were reformulated as per the reviewer’s suggestion.

Reviewer

The hypotheses H5 are now stated in a correct way, principally. The authors now rewrote the argumentation lines before which is a bit contrary to the hypotheses H5.1 and H5.2. It is not clear why the main difference should be the case as the cited references go into the direction of a moderation effect. Also, it is not clear why there should be differences in countries. There are no references except [5] about that. As the main message at page 6 is that the differences between men and women rely on moderating variables the corresponding hypotheses are not fitting. In a whole, these hypotheses now look (at a second reading) incongruous. Possibly this part should not be included as the hypotheses imply a whole research question and the main point of the study wants simply to make a validation study of the SIJS.

This point also regards the discussion.

Reviewer

2.a, now additionally

After reading the introduction again in the light of the other comments I have to admit that this critique is important: “Reviewer 3. 5. Introduction is well-written but the rationale for selecting SIJS among other measures is weak. „

The authors write „These short multi-item measures maintain the overall perspective of the measure with the advantage of obtaining the latent measure through the necessary multiple items. [..] One of these short measures is the SIJS which has some advantages over other measures. It is freely available, has shown good psychometric properties and has been used in different cultures [41–44].“

This text sounds like as there was a presentation of the other instruments which has not been done. As there is a lot of literature available about job satisfaction in general but few about single-item measurements or facet-items or facet scales measurements of job satisfaction this aspect should be presented more clearly. To present the SIJS and argue that this has advantages implies that the presented arguments can be validated a bit better in comparison to other (here not presented) scales. This is important especially for a scientific text as it always should rely on transparency.

3 Methods

The data set does not provide information for occupation, school level, socioeconomic level. If PLOS needs the complete data set then this information should be added.

The description of the data set is not satisfying for re-analyses as e.g. the scale information is lacking (which is understandable as this is not possible in an xls-file in comparison to an SPSS-file).

The internal consistency values for the instruments for the previous versions should be included in the methods section.

Answer Authors

3 - The occupational group, academic level, and socioeconomic status were added to the dataset (i.e. S1_Dataset.csv).

Unfortunately, csv files do not allow to added labels to variables or values.

The internal consistency values for the instruments previous versions were added.

Reviewer

Ok, done partly. Besides a .csv a description of the labelling could be provide. It is not clear, e.g., whether the authors used the scale anchors between the marking points 1 and 5: “Subjects are asked to respond to each item by checking a fivepoint scale ranging from 1 - “Strongly Disagree” to 5 - “Strongly Agree” (Table 2), two of those items are reversed.”

So e.g., have the points 2, 3, 4 also been labelled for the respondents?

There should be a description of the data set. Typically this is a codebook. In SPSS this is done partly automatically via the labels etc.

4 Data and analyses

As minor point, in “Table 3. Sample 1 and 2 Descriptive Statistics.” It would be helpful to have at least an abbreviation of the item content.

Answer Authors

4 -The items’ content is presented in Table 2. However, if the reviewer finds it strictly necessary, the items content can be repeated in this table.

Reviewer

I still find it much more helpful for a reader to find abbreviations like “fairly satisfied” in Table 3 in comparison to “Item 1”.

5 Discussion

The discussion follows the argumentation line, refers to the literature and research findings and discusses the results in this context.

Answer Authors

5 - Thank you.

6 Minor points

The language should be checked. E.g. the following sentence is not fully clear: „The meaning attached to work seems that not differs significantly among countries (Judge et al., 2001), as so it expected that job satisfaction differences between countries are not so great or inexistent.“

Some other examples.

„As such, each of these three dimensions was three items“

Answer Authors

6 - We carefully reviewed the writing of the manuscript.

Reviewer

Ok.

Reviewer #3: I think the authors have made necessary revision. From next time, I recommend authors to indicate "where in the main text the revision was made."

7. PLOS authors have the option to publish the peer review history of their article (what does this mean?). If published, this will include your full peer review and any attached files.

Reviewer #1: No

Reviewer #2: Yes: Paul Jimenez

Reviewer #3: Yes: Akinori Nakata

---

## [Author Response · Author response to Decision Letter 1]

19 Mar 2020

Thank you for Reviewer #2 comments. We found the comments very helpful and made a sincere effort to revise the paper based on the received feedback. 

####################################################

Reviewer 2

####################################################

2 Introduction, hypotheses

The hypotheses H5 are now stated in a correct way, principally. The authors now rewrote the argumentation lines before which is a bit contrary to the hypotheses H5.1 and H5.2. It is not clear why the main difference should be the case as the cited references go into the direction of a moderation effect. Also, it is not clear why there should be differences in countries. There are no references except [5] about that. As the main message at page 6 is that the differences between men and women rely on moderating variables the corresponding hypotheses are not fitting. In a whole, these hypotheses now look (at a second reading) incongruous. Possibly this part should not be included as the hypotheses imply a whole research question and the main point of the study wants simply to make a validation study of the SIJS.

This point also regards the discussion.

####################################################

Authors’ response

####################################################

We rewrote the hypothesis as per the referee's request. Now the research hypotheses are in line with the Statistical Alternative Hypothesis. Additionally, the rationale behind the hypotheses was changed to make the literature sound more logical regarding the established hypotheses. However, our idea of quoting the literature was not to support a particular form of the research hypotheses, but rather to present what other researchers have found on the subject.

####################################################

Reviewer 2

####################################################

2.a, now additionally

After reading the introduction again in the light of the other comments I have to admit that this critique is important:

“Reviewer 3. 5. Introduction is well-written but the rationale for selecting SIJS among other measures is weak. „

The authors write „These short multi-item measures maintain the overall perspective of the measure with the advantage of obtaining the latent measure through the necessary multiple items. [..] One of these short measures is the SIJS which has some advantages over other measures. It is freely available, has shown good psychometric properties and has been used in different cultures [41–44].“

This text sounds like as there was a presentation of the other instruments which has not been done. As there is a lot of literature available about job satisfaction in general but few about single-item measurements or facet-items or facet scales measurements of job satisfaction this aspect should be presented more clearly. To present the SIJS and argue that this has advantages implies that the presented arguments can be validated a bit better in comparison to other (here not presented) scales. This is important especially for a scientific text as it always should rely on transparency.

####################################################

Authors’ response

####################################################

We give information about other instruments: freely available or not, length, dimensions of the referred scales (see Introduction; 3rd and 4th paragraphs).

####################################################

Reviewer 2

####################################################

3 Methods

Ok, done partly. Besides a .csv a description of the labelling could be provide. It is not clear, e.g., whether the authors used the scale anchors between the marking points 1 and 5: “Subjects are asked to respond to each item by checking a fivepoint

scale ranging from 1 - “Strongly Disagree” to 5 - “Strongly Agree” (Table 2), two of those items are reversed.”

So e.g., have the points 2, 3, 4 also been labelled for the respondents?

There should be a description of the data set. Typically this is a codebook. In SPSS this is done partly automatically via the labels etc.

####################################################

Authors’ response

####################################################

Yes, points 2, 3 and 4 were also labeled. That information is presented in Table 2, nevertheless, we completed that information in the text (see the Subheading “Short Index of Job Satisfaction (SIJS)”):

Subjects are asked to respond to each item by checking a five-point scale (1 - “Strongly Disagree”, 2 - “Disagree”, 3 – “Undecided”, 4 – “Agree”, 5 - “Strongly Agree”) two of those items are reversed (Table 2).

The CSV file “S2_Dataset.csv” was added. All the variables have a description, the inverted items have that information of their description.

####################################################

Reviewer 2

####################################################

4 Data and analyses

I still find it much more helpful for a reader to find abbreviations like “fairly satisfied” in Table 3 in comparison to “Item 1”.

####################################################

Authors’ response

####################################################

We added a small item vignette to each of them. We also emphasized that for items 3 and 5 the presented descriptive statistics (Table 3) are presented for the recoded scores:

“Note. R – reversed items; the descriptive statistics for such items refer to the recoded scores.”

---

## [Editor Report · Decision Letter 2]

25 Mar 2020

Short Index of Job Satisfaction: Validity Evidence from Portugal and Brazil

PONE-D-19-28151R2

Dear Dr. Marôco,

We are pleased to inform you that your manuscript has been judged scientifically suitable for publication and will be formally accepted for publication once it complies with all outstanding technical requirements.

With kind regards,

Sergio A. Useche, Ph.D.

Academic Editor

PLOS ONE

Additional Editor Comments (optional):

Thanks for submitting the minor changes requested by one of the reviewers. After a careful revision of them, I am happy with the job done by the authors and their responses, and consider the article is publishable in PLOS ONE.
---

## [Editor Report · Acceptance letter]

31 Mar 2020

PONE-D-19-28151R2 

Short Index of Job Satisfaction: Validity Evidence from Portugal and Brazil 

Dear Dr. Marôco:

I am pleased to inform you that your manuscript has been deemed suitable for publication in PLOS ONE. Congratulations! Your manuscript is now with our production department. 

With kind regards,

on behalf of

Dr. Sergio A. Useche 

Academic Editor

PLOS ONE